# Clonal diversity predicts persistence of SARS-CoV-2 epitope-specific T-cell response

Ksenia V. Zornikova [1,2], Alexandra Khmelevskaya[1], Savely A. Sheetikov[1,2], Dmitry O. Kiryukhin[1], Olga V. Shcherbakova[1], Aleksei Titov[1], Ivan V. Zvyagin [3,4] & Grigory A. Efimov [1✉]

T cells play a pivotal role in reducing disease severity during SARS-CoV-2 infection and formation of long-term immune memory. We studied 50 COVID-19 convalescent patients and found that T cell response was induced more frequently and persisted longer than circulating antibodies. We identified 756 clonotypes specific to nine CD8+ T cell epitopes. Some epitopes were recognized by highly similar public clonotypes. Receptors for other epitopes were extremely diverse, suggesting alternative modes of recognition. We tracked persistence of epitope-specific response and individual clonotypes for a median of eight months after infection. The number of recognized epitopes per patient and quantity of epitope-specific clonotypes decreased over time, but the studied epitopes were characterized by uneven decline in the number of specific T cells. Epitopes with more clonally diverse TCR repertoires induced more pronounced and durable responses. In contrast, the abundance of specific clonotypes in peripheral circulation had no influence on their persistence.

[1] National Medical Research Center for Hematology, Moscow, Russia. [2] Faculty of Biology, Lomonosov Moscow State University, Moscow, Russia. [3] Department of Genomics of Adaptive Immunity, Shemyakin-Ovchinnikov Institute of Bioorganic Chemistry, Moscow, Russia. [4] Center for Precision Genome Editing and Genetic Technologies for Biomedicine, Pirogov Russian National Research Medical University, Moscow, Russia. ✉email: efimov.g@blood.ru

The ongoing pandemic of Coronavirus disease 2019 (COVID-19), caused by the novel severe acute respiratory syndrome coronavirus 2 (SARS-CoV-2), has resulted in considerable morbidity and mortality worldwide. It is well established that recovery from COVID-19 is mediated not only by the production of neutralizing antibodies but also by the development of a T-cell response to SARS-CoV-2 antigens[1–3].

The levels of circulating binding and neutralizing antibodies are highly variable between individuals, but in most patients, humoral response decays over the first 6–8 months post-infection[4–6]. It has been shown that SARS-CoV-2-specific memory B cells[6–10] and T cells are maintained in COVID-19 convalescent patients at least for 6–8 months[4,6,11,12], and the latest reports further extend this margin to up to 10–15 months[10,13]. Although the numbers of circulating T cells decrease over time[5,10], these cells retain the ability to secrete cytokines[14] and to expand[15] upon re-stimulation with the appropriate antigen.

SARS-CoV-2-specific T-cell responses are more prevalent than seropositivity, and can be detected in the majority of the patients who have not developed specific antibodies[16,17]. This makes the assessment of specific T-cell reactivity an accurate indicator of past infection[18,19]. In some cases, pre-existing T-cell responses might even prevent the development of full-blown disease[19].

Mounting evidence points to the clinical importance of T cells in COVID-19. Specific CD4+ and CD8+ T-cell responses contribute to reduced severity of disease[20–22]. Early induction of a functional SARS-CoV-2-specific T-cell response is associated with rapid viral clearance and milder disease[23]. On the contrary, T-cell anergy is connected with a poor prognosis[24]. Reduced diversity of T-cell response is characteristic of patients with COVID-19 pneumonia[25], while decreased numbers of CD8+ T cells indicate a high probability of developing severe disease and even death[26].

There is evidence that T cells can successfully clear the virus even in the absence of antibodies. Patients with humoral immunodeficiency or depleted B cells successfully recover from COVID-19[27–29]. Moreover, in a large prospective study, we have recently shown that a subgroup of patients lacking a humoral response to SARS-CoV-2 was partly protected from infection by T cells[30]. A beneficial role of T cells has also been shown in mice: in the absence of antibodies, T cells were able to eliminate SARS-CoV infection, and immunization with a single immunodominant CD8+ T-cell epitope protected against lethal disease[31,32].

We and others have identified multiple CD8+ and CD4+ immunogenic SARS-CoV-2 epitopes[33–38]. Recently, we systematically characterized the immunogenicity of a panel of unique SARS-CoV-2 epitopes presented by the most common HLA alleles[19]. Some of identified epitopes were immunodominant, implying they cause immune response in ≥50% of convalescent donors[19,34]. A number of publications have reported sequences of T-cell receptors (TCRs) recognizing some immunodominant epitopes of SARS-CoV-2[38–41]. It was demonstrated that TCRs specific to some epitopes are public[38,42,43] and have a high degree of mutual similarity, which is sometimes accompanied by strong biases in variable (V) and joining (J) gene usage. According to the structural data, the latter can be explained by germline-based epitope recognition[44]. The notion of SARS-CoV-2-specific TCRs could be used to identify patients exposed to this virus by analyzing individual TCR repertoires[41,45,46].

Recurrent mutations in the Spike (S) protein have given rise to variants of concern, including the current omicron variant, which are less susceptible to neutralization by antibodies[47]. At the same time, the large numbers and variability of T-cell epitopes recognized in different individuals make T-cell response more resilient against immune evasion. Several non-synonymous mutations in

known T-cell epitopes have been reported, resulting in diminished or abrogated MHC binding or T-cell activation[44,48]. However, these mutations only seldom get fixed into variants of concern, and this is probably because variants escaping presentation by one HLA allele often become binders to other alleles[49].

One of the key remaining questions is what factors influence the longevity of SARS-CoV-2-specific T-cell response. Careful analysis of T-cell response characteristics at the level of individual epitopes and clones might unravel factors influencing the formation and persistence of long-lived T-cell memory. To this end, we have measured humoral and T-cell responses to SARS-CoV-2 in paired blood samples of 50 COVID-19 convalescent patients (CP) shortly after infection and at a median of 8 months after infection. Moreover, for a subgroup of 26 CP, we characterized the frequency of memory T cells, their clonal structure, and the TCR repertoire of the T-cell immune response to a panel of eight CD8+ epitopes. We found that T-cell response was induced more often than seropositivity and persisted in a larger group of patients. Nevertheless, cellular immunity also diminished over time, manifesting as a decrease in the number of recognized antigens and epitopes, reduced frequency of specific T cells in circulation, and a reduced number of epitope-specific clonotypes. Still, most epitopes were recognized for up to 10 months post-infection.

We have described a total of 756 epitope-specific T-cell clonotypes and demonstrated that the response to some of these epitopes was mediated by public TCRs and TCRs with highly similar CDR3β amino acid sequences, whereas others were recognized by highly diverse receptors. Most epitope-specific clonotypes were present at levels below the limit of detection in peripheral blood. The most important factor influencing the dominance of response to a particular epitope and its persistence was the number of specific clonotypes detected after infection, while the average size of the clonotypes did not have a significant impact. In the face of neutralizing antibody escape by emerging variants, cellular immunity has growing importance in terms of reducing the severity of COVID-19 infection. So far, most widely-used COVID-19 vaccines are based on the S protein; in contrast, of the nine epitopes designated as immunodominant in this work, only three are S-derived. This study provides a rationale for including immunodominant T-cell epitopes in the new generation of vaccines to ensure the formation of long-term T-cell memory.

## Results

**Higher incidence and superior persistence of cellular over humoral response**. To study the dynamics of adaptive immune response to SARS-CoV-2, we recruited a cohort of 50 convalescent patients (CPs) who had asymptomatic ($n = 2$), mild ($n = 31$), or moderate to severe ($n = 17$) COVID-19 according to the classification used by the U.S. National Institutes of Health. None of the patients required treatment in the intensive care unit, oxygen supplementation, or invasive ventilation support. The cohort included 27 females and 23 males aged 17–64 years (median = 36). Peripheral blood was collected at two time-points: between 17 and 72 days (median = 35) after disease onset (TP1), and between 180 and 292 days (median = 242) after disease onset (TP2). None of the donors had been vaccinated against COVID-19 or had confirmed reinfection between samplings. A control group of healthy donors (HD) included 19 individuals. Fourteen samples from HDs cryopreserved no later than August 2019 were obtained from the biobank. The remaining five recruited during the COVID-19 pandemic had no self-reported symptoms or positive PCR, four of them subsequently became infected, and

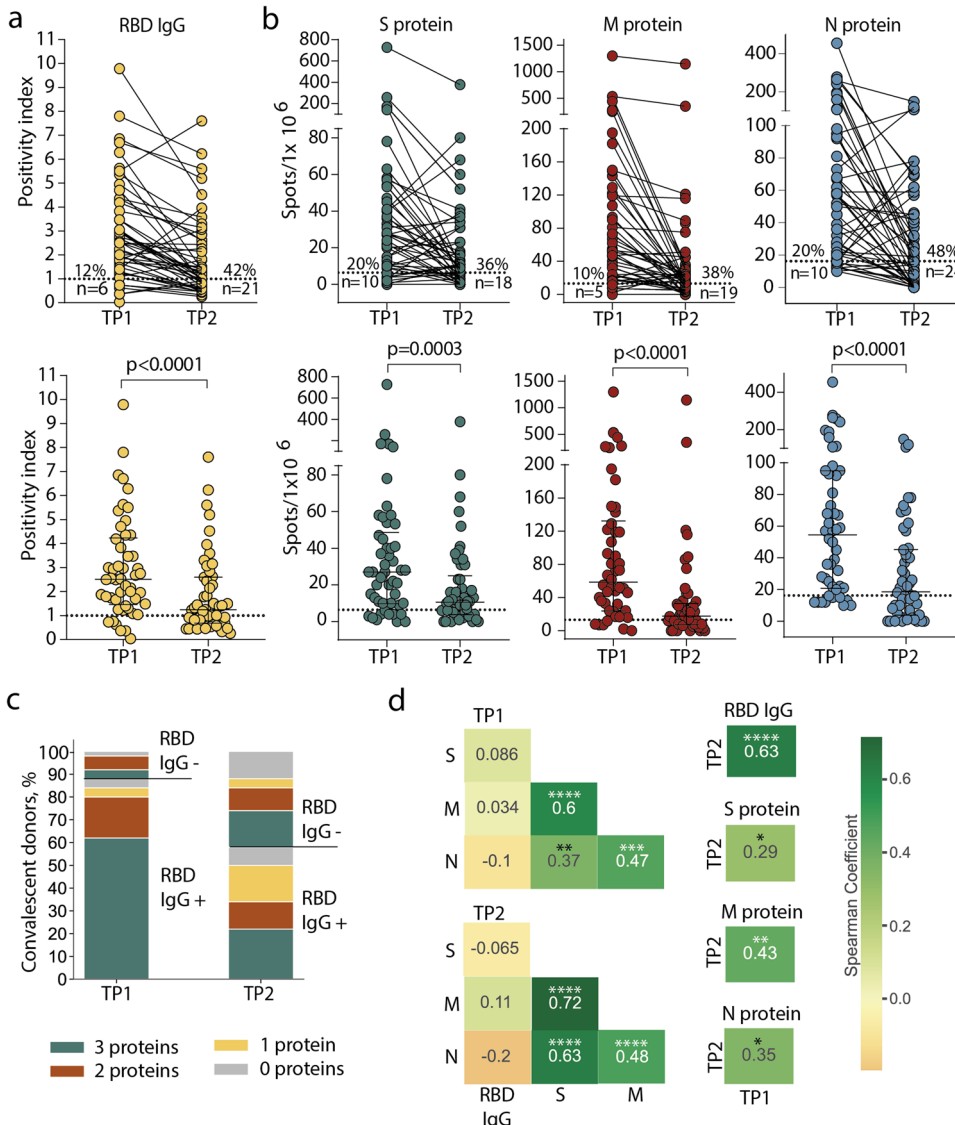

**Fig. 1 Persistence of humoral and cellular response in CP cohort. a** Levels of anti-RBD IgG as measured by ELISA at the two time-points. Means of two independent measurements are plotted. Upper plot shows samples from the same donor, connected by lines. Lower plot shows median, whiskers show interquartile range (paired Wilcoxon test); **b** magnitude of T-cell response to pools of peptides derived from S (green), M (red), and N proteins (blue) as measured by IFNγ ELISpot. Means of two independent measurements with negative control subtracted are plotted. Plots are presented as in **a**. Dotted lines in **a** and **b** mark cut-offs, the number and share of individuals without detectable response is indicated. $N = 50$ for both **a** and **b**; **c** distribution of immune responses in CPs. Colors indicate T-cell response to 0–3 peptide pools; RBD IgG+ and RBD IgG- respectively indicate the presence and absence of anti-RBD IgG; **d** Spearman correlation between humoral and cellular responses to different SARS-CoV-2 antigens. *$p \le 0.05$; **$p \le 0.01$; ***$p \le 0.001$; ****$p \le 0.0001$.

their post-infection samples were included in the CP cohort. Detailed information about the donors is provided in Supplementary Table 1.

We tested all individuals for the presence of IgGs to the receptor-binding domain (RBD) and T-cell response to pools of peptides derived from the Membrane protein (M), Nucleoprotein (N), and Spike protein (S).

A humoral response was induced in the majority of individuals: at TP1, 88% of CPs had detectable levels of anti-RBD IgG antibodies (Fig. 1a). Individual antigens demonstrated comparable levels of T-cell reactivity: 80–90% of CPs had detectable specific T cells depending on the tested proteins (Fig. 1b). However, overall T-cell reactivity was more frequently observed than humoral response: only three (6%) individuals lacked T-cell reactivity to any of the tested antigens versus six (12%)

seronegative patients. Accordingly, in five out of six seronegative individuals, we detected T-cell responses to two or three individual antigens. Of the three donors without T-cell reactivity, a humoral response was detected in two (Fig. 1c).

After eight months, the immune response to SARS-CoV-2 decreased dramatically. Only 29 (58%) CPs maintained detectable IgG levels at the TP2 (Fig. 1a). T-cell response also significantly diminished over time: depending on the tested antigen, specific T cells could not be detected at TP2 in 18–24 (36–48%) individuals, compared to 5–10 (10–20%) at TP1. N protein was characterized by the lowest number of responders at TP2 (Fig. 1b). Nevertheless, overall T-cell reactivity to any of the antigens was more persistent than antibody response: 10 (20%) individuals lacked detectable T cells compared with 21 (42%) seronegative individuals at TP2. Notably, eight of the

21 seronegative CPs had measurable T-cell responses to all three antigens (Fig. 1c). T-cell responses to the S, M, and N proteins were correlated regardless of the time point. The response to each protein correlated between two time points (Fig. 1d, Supplementary Figure 1). Antibody levels correlated between two time points and did not correlate with T-cell responses to all antigens. We did not observe any correlation between antibody titer and age or sex of donors or period of blood sampling or disease severity within either TP1 or TP2 (Supplementary Figure 2a–d,); similarly, the number of antigen-specific T cells was not considerably influenced by age or sex of donors or time since disease onset (Supplementary Figure 2e–h).

Some donors demonstrated an increase of response at TP2, including six in whom we observed an increase of T-cell responses to two or three antigens (Supplementary Figure 2i). One of the explanations is the contact with SARS-CoV-2 between samplings.

**Epitope-specific CD8+ T cells response decreases but remains detectable over eight months**. To study T-cell response at the level of individual epitopes, we selected 20 CD8+ epitopes presented by common HLA I alleles and derived from different SARS-CoV-2 proteins, and which were previously described as immunogenic (Table 1). For 15 epitopes, we successfully folded peptide-MHC (pMHC) complexes. Based on the HLA genotype, we selected a subgroup of 26 donors (Supplementary Table 2) and tested 4–15 donors (median = 8.5) for each epitope. To increase the frequency of epitope-specific memory T cells, we performed rapid in vitro antigen-specific memory cell expansion[19,38,50,51]. Epitope-specific cells were further detected by flow cytometry using MHC tetramers (Supplementary Figure 3). Each expansion was performed in triplicate, and we used the number of positive wells as a surrogate indicator of T-cell frequency.

We confirmed immunogenicity of twelve out of the 15 tested epitopes, and 11 of them were immunodominant, with epitope-specific cells detected in >50% of donors. Six epitopes (YLQ, ALW, LLY, KCY, KTF, and MEV) induced reactivity in 100% of CPs with the relevant HLA allele, and the response to four of them was detected in 3/3 wells (Fig. 2a). Only one epitope (LLY) yielded a weak response in a single HD. At TP2, T cells specific to all 12 immunogenic epitopes were still detectable in at least a share of the patients, but the number of recognized epitopes per patient decreased significantly (Fig. 2b). Number of recognized epitopes was not considerably influenced by sex or age of donors or disease severity (Supplementary Figure 4). Reduced frequency of specific T cells was characteristic of individual epitopes rather than patients: in most individuals, only a subset of the epitopes lost their immunogenicity (Fig. 2c, Supplementary Figure 5). In two donors (p1495 and p1426), we detected almost complete disappearance of cells specific to the studied epitopes, whereas, in donor p1445, these cells were more abundant at TP2.

**Dominant epitope-specific CD8+ T-cell response arises from clonal diversity**. We analyzed the repertoires of TCRβ chains of MHC-tetramer+ populations and total peripheral blood mononuclear cell (PBMC) fractions by high-throughput sequencing. Epitope-specific T-cell clonotypes were defined as sequences that were strongly (≥10-fold) and significantly ($p < 10^{-12}$, Fisher's exact test) enriched in the MHC-tetramer+ population (Fig. 3a, Supplementary Figure 6). We identified 756 clonotypes specific to nine epitopes (Supplementary Data 1). Of all the epitopes, KCY and KTF had the most diverse response, with a median of 27 clonotypes per patient for both (Fig. 3b). The epitopes characterized by the highest clonality were also those for which we observed the most robust response (Fig. 2a). The number of

**Table 1 Peptides of SARS-CoV-2 used in this study.**

| # | Peptide name | AA sequence | Length | Protein | Position, aa | Allele | Binding score | Binding rank | Reference |
|---|---|---|---|---|---|---|---|---|---|
| 1 | ALS | **ALSKGVHFV** | 9 | ORF3a | 72–80 | **A 02:01** | 0.7979 | 0.1068 | 34,36,37 |
| 2 | ALW | **ALWEIQQVV** | 9 | ORF1ab | 4099–4107 | **A 02:01** | 0.8387 | 0.0523 | 33 |
| 3 | AQF | AQFAPSASAF | 10 | N | 305–314 | B 40:01 | 0.3137 | 1.247 | 51 |
| | | | | | | B 44:03 | 0.3113 | 0.8978 | |
| 4 | ATS | **ATSRTLSYYK** | 10 | M | 171–190 | **A 03:01** | 0.7239 | 0.0566 | 33 |
| 5 | FTS | FTSDYYQLY | 9 | ORF3a | 207–215 | C 07:02 | 0.4888 | 0.1246 | 33,36,37 |
| 6 | KCY | **KCYGVSPTK** | 9 | S | 378–386 | **A 03:01** | 0.4513 | 0.9738 | 33,37 |
| 7 | KLW | **KLWAQCVQL** | 9 | ORF1ab | 27–35 | **A 02:01** | 0.7972 | 0.1083 | 33 |
| 8 | KTF | **KTFPPTEPK** | 9 | N | 361–369 | **A 03:01** | 0.7260 | 0.0548 | 33,36,37 |
| | | | | | | C 07:02 | 0.1164 | 8.268 | |
| 9 | LLLD | **LLLDRLNQL** | 9 | N | 222–230 | **A 02:01** | 0.7823 | 0.1398 | 36 |
| | | | | | | C 07:02 | 0.2790 | 1.0954 | |
| 10 | LLL | **LLLDRLNQL** | 10 | N | 221–230 | **A 02:01** | 0.5693 | 1.1237 | 34 |
| 11 | LLY | **LLYDANYFL** | 9 | ORF3a | 139–147 | **A 02:01** | 0.9193 | 0.0071 | 33,36 |
| 12 | MEV | **MEVTPSGTWL** | 10 | N | 322–331 | **B 40:01** | 0.6745 | 0.1078 | 34 |
| | | | | | | B 44:03 | 0.5156 | 0.1862 | |
| 13 | NRF | **NRFLYIIKL** | 9 | M | 43–51 | **B 27:05** | 0.56461 | 0.368 | 34,37 |
| 14 | QLR | **QLRARSVSPK** | 10 | ORF7a | 76–85 | **A 03:01** | 0.6563 | 0.1639 | 34 |
| 15 | RLQ | **RLQSLQTYV** | 9 | S | 1000–1008 | **A 02:01** | 0.7709 | 0.1610 | 38 |
| 16 | SEL | **SELVIGAVIL** | 10 | M | 136–145 | **B 40:01** | 0.7090 | 0.0713 | 34 |
| | | | | | | B 44:03 | 0.4388 | 0.3593 | |
| 17 | VVH | VVFLHVTYV | 9 | S | 1060–1068 | C 07:02 | 0.2415 | 1.665 | 36,37 |
| 18 | VYQ | VYFLQSINF | 9 | ORF3a | 112–120 | C 07:02 | 0.4441 | 0.1981 | 34,37 |
| 19 | VYI | VYIGDPAQL | 9 | ORF1ab | 5840–5848 | **A 02:01** | 0.4812 | 0.1400 | 34,36,37 |
| 20 | YLQ | **YLQPRTFLL** | 9 | S | 269–277 | C 07:02 | 0.5666 | 0.0523 | 33,36,38 |

HLA binding score and rank were predicted by NetMHCpan 4.1. M Membrane, N Nucleocapsid, S Spike. Combinations of peptides and restricting HLA alleles selected for rapid in vitro antigen-specific expansion are highlighted in bold.

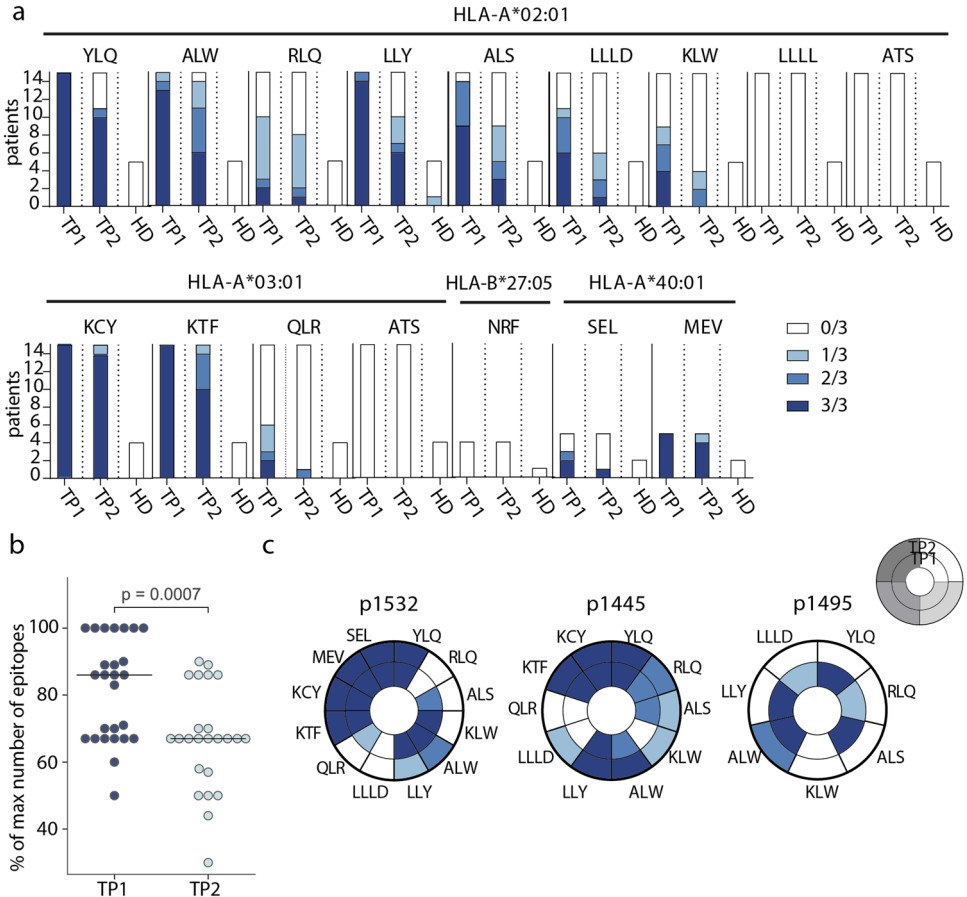

**Fig. 2 Frequency of epitope-specific T cells in CPs and healthy donors (HD). a** Number of wells with detected epitope-specific CD8+ cells after rapid in vitro expansion of PBMCs from CP ($n = 26$) and HD ($n = 7$). Bar height represents the number of HLA-allele carriers tested for response to each epitope. Color intensity indicates the number of wells containing MHC-tetramer+ cells. Epitopes represented by three or four-letter codes and their restricting HLA alleles are listed at top; **b** proportion of recognized epitopes out of the total number of tested epitopes—i.e., epitopes presented by the individual's HLA alleles. Each dot represents one donor, response in at least one well is considered positive. Paired Mann–Whitney test, with p-value indicated above; **c** changes in the detection of antigen-specific cells after rapid in vitro expansion in TP1 (inner circle) versus TP2 (outer circle) for three representative donors (p1532, p1445, p1495). Each segment corresponds to epitopes indicated by three or four-letter codes.

T-cell clonotypes specific for KCY and KTF significantly decreased in TP2 (Fig. 3b). We observed a similar reduction in diversity when we compared the response to all epitopes. The median number of clonotypes per epitope was 13 and 5 in TP1 and TP2, respectively (Fig. 3c). Nevertheless, for all epitopes except RLQ, some epitope-specific clonotypes were detectable at TP2. We did not observe any correlation between number of epitope-specific clonotypes and age or sex of donors or disease severity within either TP1 or TP2 (Supplementary Figure 7a-c). The initial disparity in the number of specific clonotypes per epitope became less prominent at TP2 (Fig. 3b). We did not find a correlation between the diversity of epitope-specific responses at the two time-points (Supplementary Figure 7d). The number of clonotypes decreased at TP2 even for the epitopes that produced a response in 3/3 wells (Supplementary Figure 7e). Nevertheless, diverse clonal structure was associated with better survival of epitope-specific T-cell response; epitopes with more stable response (i.e., the same number of wells in which MHC-tetramer + cells were detected at both TP1 and TP2) were characterized by a higher number of epitope-specific clones at TP1 (Fig. 3d). The described epitope-specific T-cell clonotypes were either undetectable or were observed at a very low frequency in the total TCR repertoire at both time-points. Only a few clonotypes were present at a frequency of >10-4 of all clonotypes (Fig. 3e). Nevertheless, the KCY, KTF, YLQ, and ALW epitopes—for which

dominant responses were retained at TP2 (Fig. 2a)—all had more frequent specific clonotypes detectable at TP1 (Fig. 3e).

The clonal repertoire of epitope-specific CD8+ cells was markedly different between the two time-points. Of the 756 discovered epitope-specific clonotypes (including public clonotypes), only 40 (5.3%) were found at both time-points (Fig. 3f). This corresponded to 0–5 (median = 1.5) persistent clonotypes per patient. In order to rule out the possibility that this negligible overlap was the result of overly stringent criteria for designating clonotype specificity, we searched the epitope-specific clonotypes of TP1 in the total MHC-tetramer+ fraction of the TP2, and vice versa. This recovered 45 additional clonotypes, specific to various epitopes, bringing the share of persistent clonotypes to 11.2% (Supplementary Figure 7f).

There was no obvious correlation between the relative clonotype abundance in the TP1 T-cell expansion and its presence or abundance in the TP2 T-cell expansion (Fig. 3g, Supplementary Figure 8). Likewise, we did not observe a correlation between clonotype persistence and the frequency with which it appeared in the repertoire at TP1 (Supplementary Figure 8, Supplementary Table 3). Only 5.9% and 4.7% of the epitope-specific clones from TP1 and TP2, respectively, were observed in the total PBMC repertoire, and we only observed 3% overlap in clonotypes from the total PBMC repertoires at the two time-points (Fig. 3f, Supplementary Figure 7g).

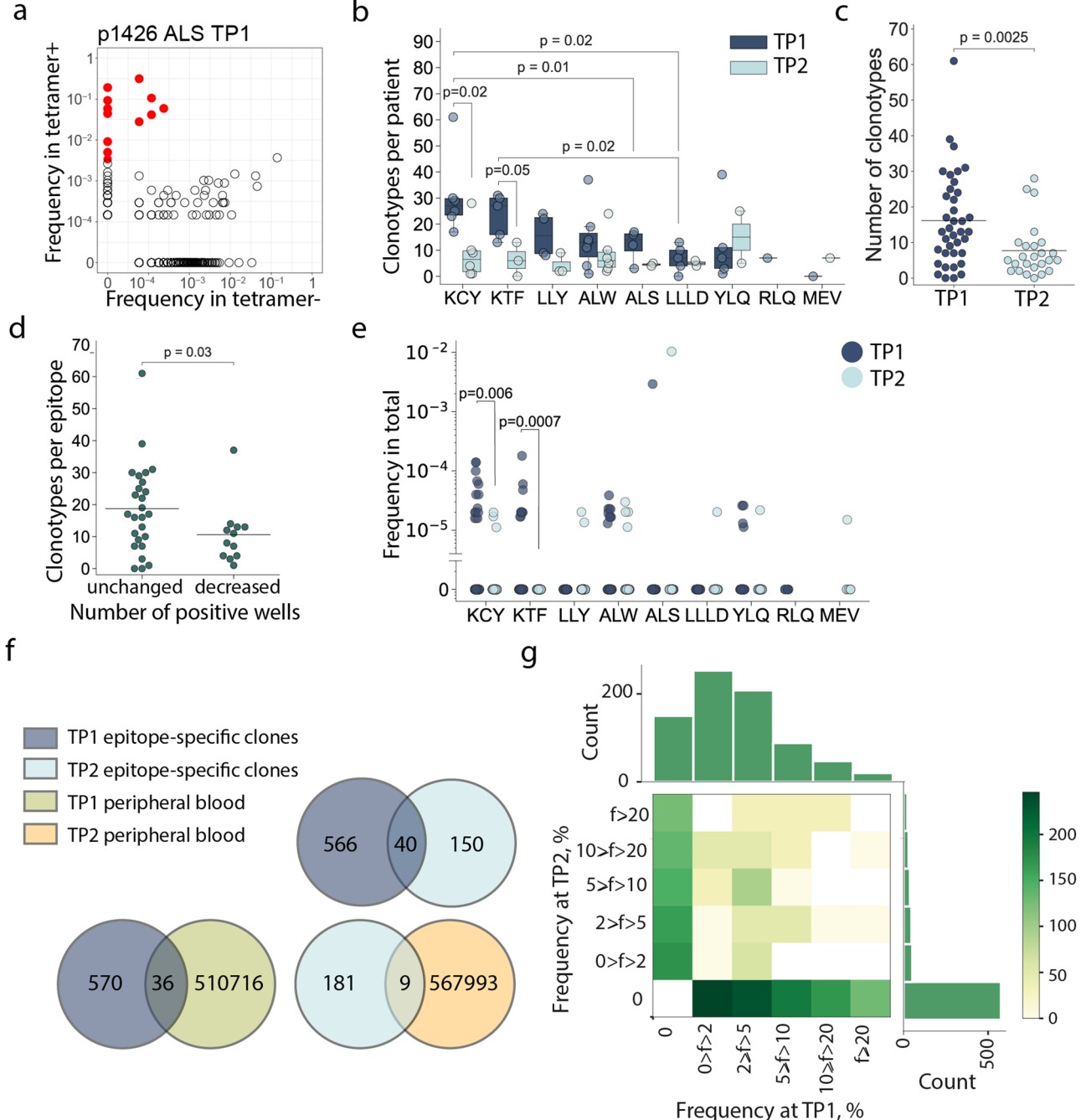

**Fig. 3 Clonal structure of the SARS-CoV-2 epitope-specific T-cell response. a** MHC-tetramer+ CD8+ cells were subjected to fluorescence-activated cell sorting (FACS) after rapid in vitro peptide-induced expansion of PBMCs from CP (n = 14), and their TCRβ repertoire was sequenced. A representative enrichment plot is shown for patient p1426, epitope ALS at TP1. Frequencies of CDR3β sequences in the MHC-tetramer⁻ and the MHC-tetramer+ populations are plotted. Red dots represent clonotypes that are strongly (>10-fold) and significantly (p < 10⁻¹², Fisher's exact test) enriched in the MHC-tetramer+ population; **b** numbers of specific clonotypes for each epitope at TP1 (dark blue, n = 36) and TP2 (light blue, n = 25). Boxplots show the first and third quartiles with median; whiskers show 1.5 interquartile range values; **c** number of epitope-specific clonotypes in the CP cohort, where each dot represents a unique patient-epitope combination (medians are shown). TP1 - dark blue, n = 36 and TP2 - light blue, n = 25; **d** number of specific clonotypes per epitope at TP1 which exhibited an unchanged (n = 26) or decreased (n = 13) overall response at TP2; **e** frequency of epitope-specific clonotypes in PBMC at TP1 (dark blue) and TP2 (light blue); **f** Venn diagrams showing overlap between different clonotype fractions from the CP cohort; **g** frequency (**f**) of epitope-specific clonotypes in expansions performed at the two time-points. Color represents the number of clonotypes in each bin. Mann–Whitney p values are shown only where significance was reached.

**The persistence of epitope-specific T cells correlates with response diversity.** Out of the seven epitopes for which TCRs were obtained from more than one donor, LLY and YLQ demonstrated the highest similarity in their epitope-specific CDR3β as measured by the share of sequences belonging to clusters formed by the sequences with one or two amino acid substitutions (Fig. 4a, b). When only a single amino acid substitution or indel was permitted, 59.7% of the LLY-specific clonotypes were assigned to a cluster, versus 10.7% and 6.2% for KTF and KCY-specific clonotypes, respectively, which were characterized by the lowest similarity. Of

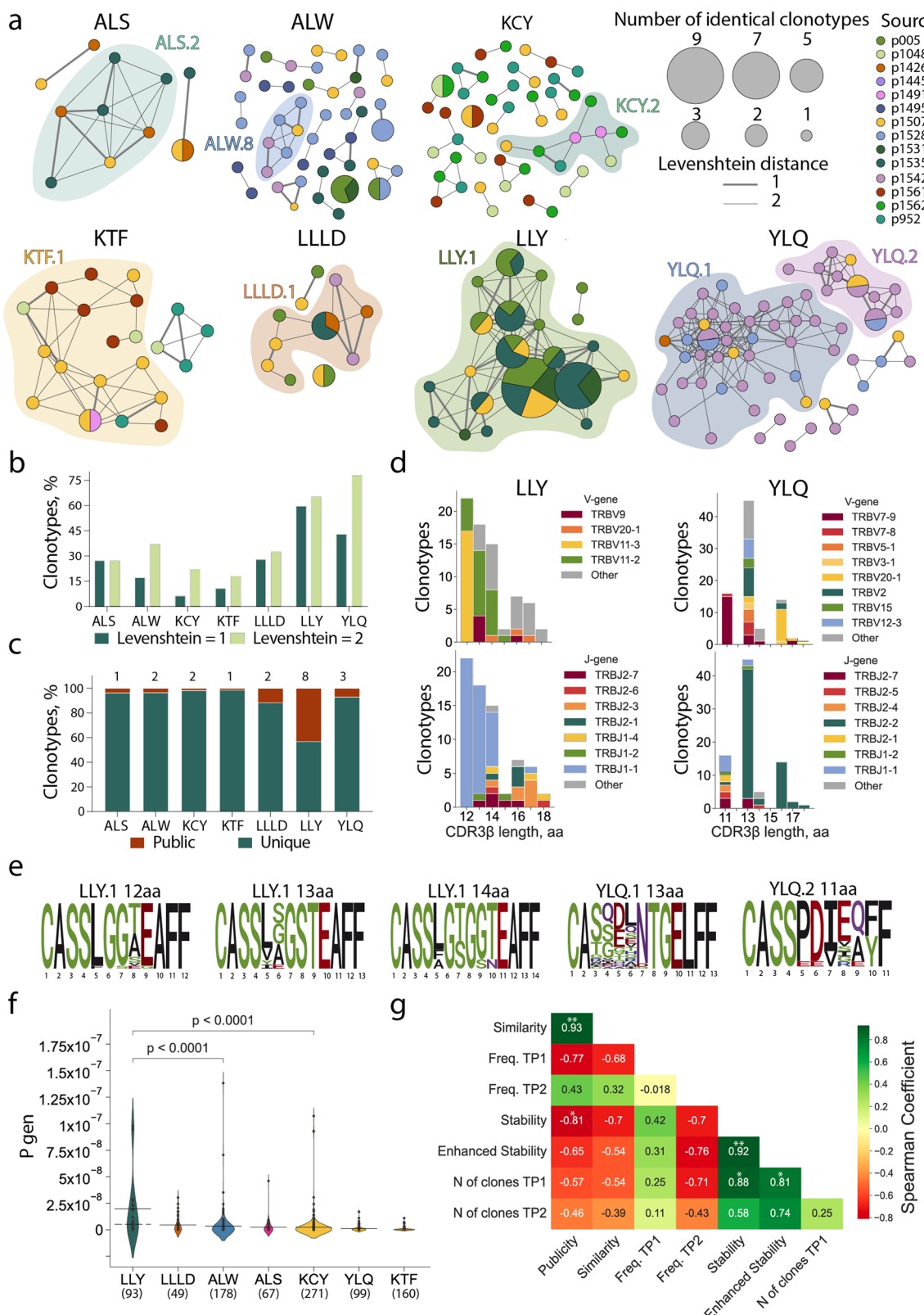

the 715 unique epitope-specific clonotypes, 19 were public, with the same CDR3β amino acid sequence shared by 2–4 individuals in our cohort (some were also encoded by multiple nucleotide sequences in one individual). Epitope LLY was recognized by the greatest number of public clonotypes—eight, corresponding to 43.1% of all LLY-specific clonotypes—while KTF and KCY had only one and two, respectively (Fig. 4c).

Next, we compared CDR3β regions with specificity for LLY, YLQ, KTF, LLLD, and ALS epitopes with CDR3β sequences annotated as recognizing the same epitopes in the Multiplex Identification of T-cell Receptor Antigen Specificity (MIRA) dataset[41] and the VDJdb database (http://vdjdb.cdr3.net)[52,53]. Both datasets contain CDR3β sequences that are highly similar to the sequences obtained in this study (Supplementary Figure 9a).

**Fig. 4 Clonal structure of SARS-CoV-2 epitope-specific response. a** Clusters of CDR3β sequences of epitope-specific CD8+ T cells. Each node represents a unique CDR3β amino-acid sequence or public sequences, the node size is proportional to the number of identical clonotypes. Lines connect similar CDR3β sequences, the thickness indicates Levenshtein distance = 1 or 2. Colors indicate donors. Only clusters with two or more members are shown. Clusters with 6 or more nodes highlighted with color; **b** proportion of clustered epitope-specific T-cell clonotypes; **c** fraction of public (dark red) and unique (dark green) epitope-specific clonotypes. Numbers of public clones are listed at top; **d** histograms of V (upper plots) and J-gene usage (lower plots) usage in LLY- (left) and YLQ-specific clones (right); **e** position-weight matrices for CDR3β sequences with the most common length observed in LLY and YLQ clusters. Cluster numbers corresponds to numbers shown in **a**; **f** probability of V(D)J-recombination with the observed V- and/or J-gene usage as calculated by the OLGA algorithm (57). Solid line represents the mean and dashed line represents the median Pgen. Mann-Whitney test, p-values are indicated, numbers of clones are listed at bottom; **g** Spearman correlation between various parameters of SARS-CoV-2 epitope-specific T-cell response. *$p \leq 0.05$, **$p \leq 0.01$, ***$p \leq 0.001$, ****$p \leq 0.0001$.

Neither MIRA nor VDJdb contained TCRs annotated as being specific for ALW or KCY epitopes. Nevertheless, we found some similar (max Levenshtein distance = 1) CDR3β sequences that were annotated as being specific for other SARS-CoV-2 epitopes (Supplementary Figure 9a). A similarity between CDR3β regions recognizing different epitopes could be explained by the impact of TCRα pairing or germline-based recognition mediated by CDR1 and CDR2. We found that 32% of the ALW-specific clonotypes described in this study were similar to YLQ-specific CDR3β sequences from the MIRA database (includes sequences specific to YLQPRTFL, YLQPRTFLL, and YVVGYLQPRTF peptides). High promiscuity was also observed for KCY, where 83 (26.9%) epitope-specific clonotypes had CDR3β sequences similar to TCRs with another specificity. Most YLQ- and KTF-specific CDR3β sequences clustered with CDR3β annotated with the same specificity in the other databases (Supplementary Figure 9a, b).

It was previously shown that epitopes of 9–10 amino acids (AA) are mostly recognized by CDR3β regions with a length of 13–15 AA. This CDR3β length is associated with public TCRs with the highest V(D)J-rearrangement probability[54]. The clonotypes specific to most of the SARS-CoV-2 epitopes obtained in this study followed this trend, with the exception of ALW and LLY. ALW-specific clonotypes had the longest CDR3β, with 16 AA being the most common length; in contrast, the majority of LLY-specific clonotypes had CDR3β of 12 AA. We did not observe YLQ-specific clones with lengths of 12 and 15 AA (Fig. 4d, Supplementary Figure 10a) because of their rarity. These lengths were observed in just 2.2% and 5%, respectively, of the 821 YLQ-specific CDR3β regions annotated in VDJdb. LLY-specific clonotypes with a short (12–14 AA) CDR3β most often used TRBV11-3 and TRBV11-2, while YLQ-clonotypes with 13 AA CDR3β were formed by a large variety of TRBVs. Clonotypes specific to both LLY and YLQ were highly biased in J-gene usage, with a predominance of TRBJ1-1 and TRBJ2-2, respectively (Fig. 4d, Supplementary Figure 10a). The high diversity of V gene usage by YLQ-specific clones is explained by the substantial role of TCRα in epitope recognition, as we demonstrated previously[44].

Position-weight matrices of the CDR3β regions with the most common lengths demonstrated more diversity in YLQ-specific than LLY-specific TCRβ repertoires (Fig. 4e). YLQ-specific clonotypes with a length of 11 (from cluster YLQ.2) or 13 AA (from cluster YLQ.1) had the same CDR3 motifs as those we described earlier[38]. LLY-specific clones with a length of 14 AA contained the CDR3 motif CASS[LFA]G[TS]G[GS][TN]EAFF were described[55], while clones with lengths of 12 and 13 AA had previously undescribed CASS[LF]G[GS][TASVE][EG]AFF and CASS[LVYI][SGAE]GSTEAFF CDR3β motifs, respectively. Moreover, we identified CDR3β motifs recognizing ALS, ALW, KCY, KTF, and LLLLD (Supplementary Figure 10b). The high similarity of the LLY-specific clonotypes and abundance of public CDR3β sequences is a result of the high probability of V(D)J-recombination (Pgen). The median Pgen as calculated by

the OLGA algorithm[56] restriction was ~$5.02 \times 10^{-9}$ (max Pgen = $9.77 \times 10^{-8}$) (Fig. 4f).

In order to elucidate the factors contributing to the longevity of response to a particular epitope, we measured the correlation of publicity (share of clones found more than one donor); similarity (share of clones belonging to a cluster); average frequency of epitope-specific clonotypes in the total repertoire of TP1 or TP2; average number of clones at TP1; and finally, stability and enhanced stability, which are respectively defined as the proportion of non-zero and 3/3 epitope-specific responses at TP2. The quantity of epitope-specific clones at TP1 strongly and significantly influenced both stability and enhanced stability (Fig. 4g). This highlights the importance of induction of polyclonal response to the formation of long-term T-cell memory.

## Discussion

We have studied the durability of humoral and cellular immune responses to SARS-CoV-2 by analyzing paired blood samples of 50 convalescent patients collected at 17–72 days and 180–292 days post-infection. Early on, T-cell response was more commonly observed than humoral response, and most seronegative individuals had measurable T-cell responses to multiple antigens. In combination with data from other studies[34,57,58], this indicates that T-cell response serves as a more reliable measure of past COVID-19 infection. After a median follow-up time of 8 months, we observed a decrease in the SARS-CoV-2-specific immune response. This is in line with other studies, where the average half-life time of antibody and T-cell response was 6–8 months and 10–15 months, respectively[4,6,10,13]. We also confirmed that the humoral response is less durable and diminishes faster. Cellular immunity against SARS-CoV-2 was present in the great majority of adults at 6 months following mild and moderate infection, as was also shown by[13]. Of the three structural antigens examined here, N protein was the least immunogenic, as shown previously[19,38,59], and the response to it was the least stable. Anti-RBD antibody levels did not correlate with T-cell response in this study, which might be characteristic of the RBD antigen. In contrast, other studies report a strong correlation between humoral response with T-cell frequencies[30,60].

The focus of this work was the analysis of CD8+ T-cell response to individual epitopes at the clonotype level. We tested the immunogenicity of 15 known epitopes in 26 donors. In order to confirm their immunogenicity, detect minor T-cell clones, and describe the structure of the antigen-specific repertoire, we used a highly sensitive method—rapid ex vivo T-cell expansion. We divided the cells equally between three wells and used the number of wells containing MHC-tetramer+ T cells after expansion as a surrogate marker of T-cell frequency. In our study, the previously reported epitopes LLLLD, ATS, and NRF were not immunogenic. The remaining epitopes differed in their immunogenicity; 11 were immunodominant with the response observed in >50% of CPs.

Only one epitope (LLY) yielded a response in a HD, which is in agreement with the other study reporting that epitope as cross-reactive[55]. At 180 to 292 days post-infection, epitope-specific T cells were still detectable, although the number of recognized epitopes decreased as well as the number of detectable epitope-specific clonotypes and their frequency. Importantly, the change in response magnitude was not universal across epitopes, with some retaining high immunogenicity for 8 months. Thus, antigen-specific T cells not only formed long-living memory populations in blood, but also proliferated after stimulation with their antigen. This finding corresponds to previous work showing the detection of SARS-CoV-1 memory T cells after 17 years[2] and the presence of SARS-CoV-2-specific memory B and T cells in the majority of a cohort of patients followed up at 6–15 months post-infection, with a reduction of the T-cell response at 12–15 months[10]. Our data confirmed that more abundant clonotypes do not generally persist better, and that clonal contraction and disappearance are more associated with a strongly proliferative phenotype[61].

We described 715 unique clonotypes. Some epitopes had high clonal diversity (e.g., KCY had 223 unique clonotypes, and KTF and 130) while others were recognized by a few clonotypes. Nevertheless, a decrease in the number of clones was characteristic even for polyclonal responses. Moreover, reduction in the number of clonotypes was shown even for high-frequency epitopes, although the clonality of the response correlated with the persistence of cells in the blood. It is worth noting that KTF and KCY were also characterized by the greatest number of specific clonotypes per patient and the most durable response. We can thus conclude that a high-diversity response contributes to persistence more than the abundance of the clones. We found very little overlap in terms of clonal representation between the two time points. This is probably because T-cell repertoires are highly dynamic and undergo major changes over the course of 6–9 months, and also because SARS-CoV-2-specific cells occupy a very small proportion of the repertoire and the probability of harvesting the same clones is low. The structure of the epitope-specific response also differed between epitopes. LLY was recognized by TCRs with the most mutual similarity, which accounted for 8 out of 19 public CDR3β sequences, while the rest of the epitopes had low levels of similarity of specific TCRs. Our work thus shows that the formation of a long-lasting immune response to SARS-CoV-2 is correlate with the clonal diversity of the initial response rather than the abundance of a particular clonotype in the peripheral circulation.

## Methods

**Study design**. 50 COVID-19 convalescent donors (CP) from Moscow, Russia volunteered to participation in this study. COVID-19 was confirmed by positive SARS-CoV-2 RT-PCR test. All donors signed the informed consent form approved by the National Research Center for Hematology ethical committee (N 150, 02.07.2020) before enrollment. The severity of the disease was defined according to the classification scheme used by the US National Institutes of Health (from www.covid19treatmentguidelines.nih.gov): asymptomatic (lack of symptoms), mild severity (fever, cough, muscle pain, but without respiratory difficulty or abnormal chest imaging) and moderate/severe (lower respiratory disease at CT scan or clinical assessment, oxygen saturation ($SaO_2$) >93% on room air, but lung infiltrates less than 50%). Additionally, 19 HD samples were obtained: 14 from the National Medical Research Center for Hematology blood bank (cryopreserved no later than August 2019) with the approval of the local ethical committee, and 5 recruited during the COVID-19 pandemic with no self-reported symptoms and negative PCR test results (four of them subsequently became infected, and their post-infection samples were included in CP cohort).

Peripheral blood of CPs was collected at two time points: between 17 and 72 days after disease onset, and between 180 and 292 days after disease onset. None of the donors had been vaccinated against COVID-19 or had confirmed reinfection between samplings.

For serology and T-cell response analysis, two replicates were performed for every experiment, and the data represent an average of the replicates with subtracted background (negative control).

**PBMC isolation**. 30 mL of venous blood from donors was collected into EDTA tubes (Sarstedt) and subjected to Ficoll (Paneco) density gradient centrifugation ($400 \times g$, 30 min). Isolated PBMCs were washed with PBS containing 2 mM EDTA and used for assays or frozen in fetal bovine serum containing 7% DMSO.

**HLA genotyping**. For most donors, HLA genotyping was performed with the One Lambda ALLType kit (Thermo Fisher Scientific), which uses multiplex PCR to amplify full HLA-A/B/C gene sequences, and from exon 2 to the 3′ UTR of the HLA-DRB1/3/4/5/DQB1 genes. Prepared libraries were run on an Illumina MiSeq sequencer using a standard flow-cell with $2 \times 150$ paired-end sequencing. Reads were analyzed using the One Lambda HLA TypeStream Visual Software (TSV), version 2.0.0.27232, and the IPD-IMGT/HLA database 3.39.0.0. Other donors were HLA genotyped by Sanger sequencing for loci HLA-A, B, C, DRB1, and DQB1 using Protrans S4 and S3 reagents. PCR products were prepared for sequencing with BigDye Terminator v1.1 (Thermo Fisher Scientific). Capillary electrophoresis was performed on a Nanophore 05 Genetic Analyzer.

**SARS-CoV-2 peptides**. Putative 20 epitopes of SARS-CoV-2 proteins were included in the analysis if they were binders (rank < 2) according to NetMHCpan 4.1[62] and immunogenic according to published data. Detailed information about the selected peptides is listed in Table 1. Peptides (at least 95% purity) were synthesized either by Peptide 2.0 or by the Shemyakin-Ovchinnikov Institute of Bioorganic Chemistry RAS. Peptides containing Cys and/or Met were diluted in a PBS/isopropanol mixture (1:1 v/v) at concentrations of up to 10–25 mM. Other peptides were diluted in DMSO (Sigma-Aldrich) at up to 30–40 mM.

**Antigen-specific T-cell expansion**. For rapid in vitro expansion, we used PBMCs of donors with 1–4 of the HLA-A*02:01, HLA-A*03:01, HLA-B*40:01, and HLA-B*27:05 alleles, as previously described[50]. Briefly, $9 \times 10^6$ cells was split between three wells and incubated for 10–12 days in RPMI 1640 culture medium supplemented with 10% normal human A/B serum, 1 mM sodium pyruvate, 25 ng/mL IL-7, 40 mg/mL IL-15, and 50 ng/mL IL-2 (Miltenyi) at a final volume of 2 ml/well. Half of the medium was replaced on days 3, 5, and 7. A mix of patient HLA-specific peptides (Table 1) of SARS-CoV-2 in DMSO or MES buffer/isopropanol mixture was added on day 0. The final concentration of each peptide in the medium was 10 ng/mL.

**Production of biotinylated MHC class I/peptide complexes**. All chemicals used were of analytical grade and were purchased from Sigma-Aldrich except for EDTA, sodium azide (Amresco), sodium chloride (Malinovoe Ozero) and protease inhibitor cocktail (Thermo). Recombinant biotin ligase used was homemade. Solutions were prepared on deionized water (Simplicity Water purification system, Merck-Millipore) and filtered using 0.45 μm syringe filters (Sarstedt) or bottle top filters (Nalgene Nunc International).

Soluble HLA-A2 loaded with different peptides was prepared by in vitro folding of *E. coli* inclusion bodies based on previously published method[63] with modifications[38]. Human heavy (HLA-A*02:01 with biotinylation tag) and light (beta-2 microglobulin) chains were expressed in E. coli strain BL21(DE3) pLysS as inclusion bodies and used for in vitro folding. Peptides and light and heavy chains were mixed in folding buffer (100 mM Tris-HCl, 400 mM arginine, 5 mM reduced glutathione, 0.5 mM oxidized glutathione, 2 mM EDTA, protease inhibitors, 1 mM PMSF, pH = 8.0) at a 30:4:3 final molar ratio. Folding reactions were incubated at 8 °C for up to 5 days. Correctly-folded complexes were purified on a Superdex 75 pg 16/600 column (Cytiva) using Tris-buffered saline (20 mM Tris-HCl, 150 mM NaCl, pH 8.0) as mobile phase. Complexes were biotinylated by in-house-made biotin ligase (20 mM Tris-HCl, 150 mM NaCl, 40 mM ATP, 0.4 mM biotin, 6.5 mM MgCl2, 25 μg/ml biotin ligase, protease inhibitor cocktail) at 30 °C for 1 hour or at 8 °C overnight and purified on a Superdex 75 pg 10/300 column. Biotinylated monomers were concentrated to a final concentration of 0.4–1.0 mg/ml and stored in 20% glycerol, 0.1% sodium azide, 0.1 mM EDTA, and protease inhibitor cocktail. Concentrations were determined using specific absorbance, with A280 = 2.36 and 1.68 for HLA-A and hB2M, respectively (as calculated in SnapGene Viewer based on amino-acid sequence). Plasmids encoding HLA and b2-microglobulin were kindly provided by Ton Schumacher (The Netherlands Cancer Institute, Amsterdam, Netherlands).

**MHC-tetramer staining**. Antigen-specific cells were detected by staining with CD3-AF700, CD8-FITC, 7AAD, and with combinations of two different peptide-MHC-tetramer complexes conjugated with streptavidin-allophycocyanin and streptavidin-R-phycoerythrin (Thermo Fisher Scientific) as previously described[64]. We considered a sample well as positive if the percent of MHC-tetramer+ CD3+ CD8+ cells was >0.03–0.4%, depending on the MHC-tetramer. An Aria III cell sorter (BD Biosciences) was used to sort cells, and data were analyzed using FlowJo Software (version 10.6.1).

**ELISA**. We used the IVD ELISA kit (National Research Center for Hematology) for the detection of anti-RBD IgG according to the manufacturer's instructions. The optical density (OD) was measured at 450 nm with a reference of 650 nm on a

MultiScan FC (Thermo Fisher Scientific) instrument. The mean of two OD values for each sample was divided by the mean of two OD values of positive control and used as a positivity index. The cutoff was determined according to manufacturers' instructions, and all samples with a positivity index >1 were considered positive.

**IFNγ ELISPOT.** IFNγ production by antigen-specific T cells was measured with the ImmunoSpot human IFNγ single-color ELISPOT kit (CTL) with a 96-well nitrocellulose plate precoated with the human IFNγ capture antibody. Thawed PBMCs were plated at a density of $5 \times 10^5$ cells/well and pulsed with SARS-CoV-2 S, M, or N protein-derived peptide pools (130-126-701, 130-126-703, 130-126-699, Miltenyi Biotec) separately in duplicates at a final concentration of 1 μM in serum-free testing medium (CTL) containing 1 mM GlutaMAX (GIBCO) at a final volume 200 μL/well. Plates were incubated for 16 h at 37 °C in 5% CO2. Assays were performed according to the manufacturer's instructions. Spots were counted by CTL ImmunoSpot Analyzer using ImmunoSpot software. We subtracted the background (negative control) from each value and used the average value of two wells with the same peptide pool. We measured the T-cell response of 11 HD to determine a cut-off for each peptide pool as mean +1.69 SD.

**TCR repertoire sequencing.** TCR libraries of MHC-tetramer+ and MHC-tetramer- fractions were prepared as described previously[65]. RNA was isolated from Trizol reagent (Thermo Fisher Scientific) using Phasemaker Tubes (Thermo Fisher Scientific), the cDNA synthesis reaction for TCR β chains was carried out with a primer to the C-terminal region and SMART-Mk, providing a 5′ template-switch effect and containing a sample barcode for contamination control as well as a unique molecular identifier. TCR libraries of total PBMC samples were generated using the human multiplex TCR Kit (MiLaboratories) according to the manufacturer's instructions. Sequencing was performed with an Illumina MiSeq or NextSeq platform. TCR repertoire data were analyzed using MIXCR[66], MIGEC[67], and VDJtools software[68] with default settings.

**TCR repertoire analysis.** TCRs specific to SARS-CoV-2 epitopes were defined as clonotypes with frequencies significantly enriched in the MHC-tetramer+ fraction relative to the MHC-tetramer- fraction (≥10-fold higher frequency, $p$ value $<10^{-12}$, exact Fisher test). Clones with multi-epitope specificity were removed from the analysis. Epitope-specific TCR sequences were matched against VDJdb and ImmunoCODE datasets using the VDJmatch tool, with a maximum Levenshtein distance of 1. Graphs were plotted using "igraph" R package version 1.2.6. TCR logos were plotted using "ggseqlogo" package version 0.1.

**Statistics and reproducibility.** All data comparisons were performed using GraphPad Prism version 8.0 software and Python (version 3.9.1). For comparisons between time points, paired Wilcoxon test was performed. Peptides' HLA binding affinity score and rank were predicted by NetMHCpan 4.1. Epitope-specific TCR sequences were analyzed in R (version 4.0.0) and Python (version 3.9.1).

**Reporting summary.** Further information on research design is available in the Nature Portfolio Reporting Summary linked to this article.

## Data availability

All data are available in the main text or the supplementary materials. All numerical raw data are provided as Supplementary Data 2. Raw sequencing data are deposited to the EMBL Nucleotide Sequence Database (ENA) under accession PRJEB57236.

## Code availability

R markdown notebooks and Python file are used for data analysis are available at https://github.com/LabTransplantImmunology/tcr-seq-pipline/

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

## Acknowledgements

We would like to express our gratitude to all donors who volunteered for our study; to the nurses who performed the venipuncture (Yulia Fadeeva, Anastasya Nisanova, Valentyna Mirponova, and Lubov Piskunova); and to our colleagues Naina Shakirova and Ekaterina Khamaganova for their kind help with the experiments. We thank Michael Eisenstein for careful and thoughtful manuscript editing. This work was funded by Russian Science Foundation grant 20-15-00395 (G.A.E.), Ministry of Science and Higher Education of the Russian Federation grant No. 075-15-2019-1789 (I.Z.).

## Author contributions

Conceptualization: G.A.E., K.V.Z. Methodology: G.A.E., K.V.Z., I.Z. Investigation: K.V.Z., A.K., S.A.S., A.T. Resources: K.V.Z., S.A.S., D.K., O.V.S. Visualization: K.V.Z., A.K. Funding acquisition: G.A.E. Project administration: G.A.E. Writing—original draft: K.V.Z. Writing—review & editing: G.A.E., K.V.Z., A.K., S.A.S., I.Z.

## Competing interests

The authors declare no competing interests.
