## [Peer Review File · Communications Biology]

Reviewers' comments:

Reviewer #1 (Remarks to the Author):

This study reports bulk (i.e., single-chain beta) TCR sequencing and analysis of 50 subjects previously infected with SARS-COV-2 and 19 control subjects with no evidence of previous infection, finding that TCR diversity correlates with T cell persistence. The size of the cohort and stretch in time after infection under study are both notable. Although the authors utilize fairly simple analytical approaches, contrasting with the recent movement in the field towards almost exclusive use of statistical approaches like ALICE (Pogorelyy et al 2019 PLOS Bio), GLIPH (Glanville et al 2017 Nature), and TCRdist (Dash et al 2017 Nature), their major findings seem to be both well-considered and robust. The data and analyses presented are important, relevant to the growing literature in this field, and a good fit for this journal; however, I have a few concerns that should be addressed prior to acceptance.

1. The title should be changed from "Clonal diversity determines persistence..." to either "Clonal diversity predicts persistence..." or "Clonal diversity correlates with persistence...", as the word "determines" is suggestive of a causal correlation that could only be proven by experiments that somehow vary clonal diversity across subjects.
2. The authors suggest that the results of their study provide "a rationale for including [non-spike] immunodominant T cell epitopes in the new generation of vaccines to ensure the formation of long-term T cell memory." Frankly, other studies have already demonstrated that non-spike, immunodominant T cell epitopes arise after SARS-COV-2 infection and persist up to a year (e.g., Cohen et al 2021 Cell Rep Med; the review by Paul Moss in Nature Immunology from Feb 2022 might also be useful), and those studies also phenotypically interrogate those cells (something not done in the current manuscript). For the authors to suggest that the strength of this paper is its impact on vaccine development is not only disingenuous, but it detracts from the important work that the paper puts forward on its own merits. Ultimately, I would want to see functional assessments of epitope-specific cells >=9 months after infection to support the statement made by the authors, but I believe those experiments would be well outside the scope of the current manuscript.
3. The authors note that a key question in the field is "what factors influence the longevity of SARS-CoV-2 specific T cell responses". Their data show that the diversity of the TCR repertoire predicts persistence of that response, which is both interesting and important. The authors also investigate impacts of age and sex. But a key missing consideration is disease severity, and the authors should report whether or not there was an association there.
4. The authors seem to identify immunodominant epitopes as those with detectable responses in >50% of survey individuals. Although not traditional, it is an acceptable definition for immunodominance, but the authors should consider citing a paper that advocates for this definition. Furthermore, the authors should define it earlier in their paper, as they reference the term in the introduction but do not fully define it until Results Section 2.
5. A number of the cited references link to preprint servers rather than the peer-reviewed versions of those works; the authors should update references that have gone through peer review, and importantly the authors should confirm that the relevant points they are referencing from those publications are still supported in the manuscripts that have completed peer review, as manuscripts often change dramatically between preprint and peer-review.
6. Figure S2D, M protein age association—is the p-value for TP2 correct? It does not look like a significant correlation, and in the text the authors suggest it is not significant.
7. Figure 2A, N protein: There are a number of patients with TP1 samples but no apparent TP2 samples. I'm curious as to why unmatched samples were included.
8. Line 179: I believe the reference to S3 should be for S4.

9. 179-191: "In two donors (p1495 and p1426), we detected almost complete disappearance of antigen-specific cells, whereas in donor p1445, these cells were more abundant at TP2." Inaccurate to say that "antigen-specific cells were more abundant at TP2" when it only appears to be the case for a single antigen. Unless I have misinterpreted here, this is an overly broad statement that requires more finesse.

10. Fig 3A: The authors should make it clear what exactly they are plotting, as they do not include a sufficiently clear explanation in the legend, and the axes are not appropriately labeled. I believe they are plotting clonotype frequencies. Assuming I'm correct, I was confused as to why the same clonotype would have both a tetramer-negative and a tetramer-positive clonotype frequency. I then assumed that it was possible that the Beta chain could be present in cells with distinct Alpha chains, and therefore the tetramer-negative cells are actually distinct (paired chained) clonotypes. However, the authors go on to speculate that clonotypes found in the tetramer-positive fraction of one time point but found in the tetramer-negative fraction of another time point instead represent technological limitations of tetramer staining and flow cytometry (I am admittedly not an expert in flow cytometry, so I cannot comment on whether this is reasonable). Do the authors believe that the single-chain limitations of the analyses could also be a contributing factor? Regardless of the potential cause of this error, I am willing to assume that the level of error is likely acceptable; however, this warrants a thorough discussion by the authors. It would be important to know if there are any other studies looking at single-chain repertoire in tetramer-positive and tetramer-negative populations simultaneously, if they see similar levels of "error", and what they attribute that error to. If we do not see this pattern in paired-chain analyses, perhaps that points to a limitation of bulk TCR sequencing rather than issues with tetramer staining and sorting.

11. The more I look at 3A, the more I like it (assuming I'm interpreting it correctly). It would likely be worthwhile to show the same type of figure for all time points and epitopes as a supplementary figure.

12. Lines 236-230: Worthwhile noting if these "tetramer-negative" clonotypes were enriched for any particular epitope, or if the spread was generally even.

13. Fig S5C and S5D are not called in order in the text.

14. Fig 4A legend, I believe "nods" should be "nodes".

15. Line 306: Reporting a mean is generally useless without also reporting the standard deviation. The authors may instead prefer to report the median.

16. A number of tools used for processing the repertoire data (MIXCR, MIGEC, VDJtools) lacked appropriate citation of the associated papers. I assume that this was an oversight, as some of the authors are also authors on these missing references, but regardless this is unacceptable and must be resolved.

Reviewer #2 (Remarks to the Author):

The authors studied 50 COVID-19 convalescent patients. They found that SARS-CoV-2-specific T cell responses were induced more often and persisted longer than SARS-CoV-2-specific antibody responses in blood. They characterized the clonotypes of CD8+ T cells specific to 9 SARS-CoV-2 epitopes and their recognition by public clonotypes. Furthermore, they tracked persistence of these epitope-specific T cells.

Overall, this is a timely, interesting and well-presented manuscript that should be of immediate interest to the readers of Communications Biology.

We thank the Reviewers for their thorough examination of the work and for their valuable comments and suggestions. The manuscript text, figures, references, and supplementary materials have been edited to address the raised issues. A point-by-point response to the Reviewers' comments is presented below. Each question or issue posed by the Reviewer is restated in italics with a reply marked with R - Response.

Response to Reviewer #1:

“This study reports bulk (i.e., single-chain beta) TCR sequencing and analysis of 50 subjects previously infected with SARS-COV-2 and 19 control subjects with no evidence of previous infection, finding that TCR diversity correlates with T cell persistence. The size of the cohort and stretch in time after infection under study are both notable. Although the authors utilize fairly simple analytical approaches, contrasting with the recent movement in the field towards almost exclusive use of statistical approaches like ALICE (Pogorelyy et al 2019 PLOS Bio), GLIPH (Glanville et al 2017 Nature), and TCRdist (Dash et al 2017 Nature), their major findings seem to be both well-considered and robust. The data and analyses presented are important, relevant to the growing literature in this field, and a good fit for this journal; however, I have a few concerns that should be addressed prior to acceptance.”

R: We are grateful to the Reviewer for a positive evaluation of our work.

“1. The title should be changed from “Clonal diversity determines persistence...” to either “Clonal diversity predicts persistence...” or “Clonal diversity correlates with persistence...”, as the word “determines” is suggestive of a causal correlation that could only be proven by experiments that somehow vary clonal diversity across subjects.”

R: We agree with the Reviewer and now change the title to “Clonal diversity predicts persistence of SARS-CoV-2 epitope-specific T cell response”.

“2. The authors suggest that the results of their study provide “a rationale for including [non-spike] immunodominant T cell epitopes in the new generation of vaccines to ensure the formation of long-term T cell memory.” Frankly, other studies have already demonstrated that non-spike, immunodominant T cell epitopes arise after SARS-COV-2 infection and persist up to a year (e.g., Cohen et al 2021 Cell Rep Med; the review by Paul Moss in Nature Immunology from Feb 2022 might also be useful), and those studies also phenotypically interrogate those cells (something not done in the current manuscript). For the authors to suggest that the strength of this paper is its impact on vaccine development is not only disingenuous, but it detracts from the important work that the paper puts forward on its own merits. Ultimately, I would want to see functional assessments of epitope-specific cells ≥ 9 months after infection to support the statement made by the authors, but I believe those experiments would be well outside the scope of the current manuscript.”

R: We agree with the Reviewer's statement and remove mention of the vaccine design from the text.

"3. The authors note that a key question in the field is "what factors influence the longevity of SARS-CoV-2 specific T cell responses". Their data show that the diversity of the TCR repertoire predicts persistence of that response, which is both interesting and important. The authors also investigate impacts of age and sex. But a key missing consideration is disease severity, and the authors should report whether or not there was an association there."

R: As the Reviewer requested, in the amended version of the paper, we have added Supplementary Figure S2D and Figure S2H, showing a correlation between immune response and disease severity.

"4. The authors seem to identify immunodominant epitopes as those with detectable responses in >50% of survey individuals. Although not traditional, it is an acceptable definition for immunodominance, but the authors should consider citing a paper that advocates for this definition. Furthermore, the authors should define it earlier in their paper, as they reference the term in the introduction but do not fully define it until Results Section 2."

R: As the Reviewer requested, we have added immunodominant epitope definition based on other papers (lines 60-62).

"5. A number of the cited references link to preprint servers rather than the peer-reviewed versions of those works; the authors should update references that have gone through peer review, and importantly the authors should confirm that the relevant points they are referencing from those publications are still supported in the manuscripts that have completed peer review, as manuscripts often change dramatically between preprint and peer-review."

R: References have been updated (refs 9, 11, 19, 30, 40 and 55).

"6. Figure S2D, M protein age association—is the p-value for TP2 correct? It does not look like a significant correlation, and in the text the authors suggest it is not significant."

R: p-value for the association between response to M protein at TP2 and age provided on the Fig. S2E is correct, but the Spearman correlation coefficient is relatively low ($r = 0.341$), and given the absence of significant correlation for the other antigens, we assume that this correlation is not biologically meaningful. To avoid any ambiguity, we have substituted significantly for considerably.

“7. Figure 2A, N protein: There are a number of patients with TP1 samples but no apparent TP2 samples. I’m curious as to why unmatched samples were included.”

R: We believe that the Reviewer refers to Figure 1B, N protein. All included samples were paired, connecting lines on this plot were deleted due to a mistake. We thank the Reviewer for noticing this, and we replaced this plot with the correct one.

“8. Line 179: I believe the reference to S3 should be for S4.”

R: We are thankful to the Reviewer for noticing this. The reference has been edited according to the Reviewer's comment.

“9. 179-191: “In two donors (p1495 and p1426), we detected almost complete disappearance of antigen-specific cells, whereas in donor p1445, these cells were more abundant at TP2.” Inaccurate to say that “antigen-specific cells were more abundant at TP2” when it only appears to be the case for a single antigen. Unless I have misinterpreted here, this is an overly broad statement that requires more finesse.”

R: We agree with the Reviewer and clarify this point in the manuscript in accordance with the Reviewer's comment (line 181).

“10. Fig 3A: The authors should make it clear what exactly they are plotting, as they do not include a sufficiently clear explanation in the legend, and the axes are not appropriately labeled. I believe they are plotting clonotype frequencies. Assuming I’m correct, I was confused as to why the same clonotype would have both a tetramer-negative and a tetramer-positive clonotype frequency. I then assumed that it was possible that the Beta chain could be present in cells with distinct Alpha chains, and therefore the tetramer-negative cells are actually distinct (paired chained) clonotypes. However, the authors go on to speculate that clonotypes found in the tetramer-positive fraction of one time point but found in the tetramer-negative fraction of another time point instead represent technological limitations of tetramer staining and flow cytometry (I am admittedly not an expert in flow cytometry, so I cannot comment on whether this is reasonable). Do the authors believe that the single-chain limitations of the analyses could also be a contributing factor? Regardless of the potential cause of this error, I am willing to assume that the level of error is likely acceptable; however, this warrants a thorough discussion by the authors. It would be important to know if there are any other studies looking at single-chain repertoire in tetramer-positive and tetramer-negative populations simultaneously, if they see similar levels of “error”, and what they attribute that error to. If we do not see this pattern in

paired-chain analyses, perhaps that points to a limitation of bulk TCR sequencing rather than issues with tetramer staining and sorting.”

R: Yes, the frequencies of clonotypes in tetramer-negative and tetramer-positive fractions are plotted here. The labels on the axes have been changed, and the explanation was added to the legend to make it clear. Due to the limitation of flow cytometry and tetramer staining (e.g. non-specific tetramer staining or incomplete staining of T cell clones low-affinity TCRs) we additionally used statistical criteria and defined epitope-specific clonotypes as sequences that were found in tetramer-positive fraction at least ten times more often than in tetramer-negative fraction. These limitations are intrinsic to flow cytometry and non-related to the bulk sequencing approach used here same limitations apply for single-cell sequencing. For example, see figures 2A and 2B in doi: 10.1038/s41590-022-01184-4).

“11. The more I look at 3A, the more I like it (assuming I’m interpreting it correctly). It would likely be worthwhile to show the same type of figure for all time points and epitopes as a supplementary figure.”

R: We thank the reviewer and add Supplementary Figure S5 showing enrichment plots for all time points and epitopes.

“12. Lines 236-230: Worthwhile noting if these “tetramer-negative” clonotypes were enriched for any particular epitope, or if the spread was generally even.”

R: According to the reviewer's comment, we clarified that the recovered paired clonotypes were specific to multiple epitopes (lines 230-231).

“13. Fig S5C and S5D are not called in order in the text.”

R: The order of C and D plots has been rearranged in the text, Figure, and legend. Due to adding of the new Fig.S5 this Figure is now referred to as Fig.S6.

“14. Fig 4A legend, I believe “nods” should be “nodes”.”

R: We are thankful to the reviewer for noticing that mistake. It has been corrected (line 323).

“15. Line 306: Reporting a mean is generally useless without also reporting the standard deviation. The authors may instead prefer to report the median.”

R: As the Reviewer requested, we report the median instead of the mean (lines 306-307).

“16. A number of tools used for processing the repertoire data (MIXCR, MIGEC, VDJtools) lacked appropriate citation of the associated papers. I assume that this was an oversight, as some of the authors are also authors on these missing references, but regardless this is unacceptable and must be resolved.”

R: The references 66 (MIXCR tool, doi: 10.1038/nmeth.3364), 67 (MIGEC tool, doi: 10.1038/nmeth.2960) and 68 (VDJtools tool, doi: 10.1371/journal.pcbi.1004503) have been added (lines 502-503 and lines 683-688).

Response to Reviewer #2:

“The authors studied 50 COVID-19 convalescent patients. They found that SARS-CoV-2-specific T cell responses were induced more often and persisted longer than SARS-CoV-2-specific antibody responses in blood. They characterized the clonotypes of CD8+ T cells specific to 9 SARS-CoV-2 epitopes and their recognition by public clonotypes. Furthermore, they tracked persistence of these epitope-specific T cells.

Overall, this is a timely, interesting and well-presented manuscript that should be of immediate interest to the readers of Communications Biology.”

R: We appreciate a positive evaluation of our work.

Reviewers' comments:

Reviewer #1 (Remarks to the Author):

In their revision, the authors have responded to each of my requests. However, I fear that one point I raised in particular was not understood, and I have attempted to clarify below.

In response to point 3, the authors include new analyses comparing antibody levels and T cell magnitude across disease severity levels. (Incidentally, please indicate results of statistical testing for S2D and S2H as in S2C). While this is useful, I was specifically requesting an analysis of how diversity of the TCR repertoire might vary as a function of disease severity (and age and sex). The goal is to determine if age, sex, or disease severity correlate at all with TCR repertoire diversity and/or epitope-specific persistence. I feel that this is important to include regardless of the result of these tests; if for instance the correlation between N clones and stability is different across ages or disease severity, this would be an important finding.

In addition, the authors' change in response to point 11 is very vague to the point of potential misinterpretation. Please simply report the fraction of these clonotypes that were associated with each epitope.

Lastly, under "Subhead 13: Data availability", the authors only list availability of previously published and publicly available analytical software. It is vital that the actual data for all of these analyses also be made publicly available upon publication.

We thank the Reviewers for their thorough examination of the work and for their valuable comments and suggestions. The manuscript text, figures, and supplementary materials have been edited to address the raised issues. A point-by-point response to the Reviewers' comments is presented below. Each question or issue posed by the Reviewer is restated in italics with a reply marked with R - Response.

Response to Reviewer #1:

“1. In response to point 3, the authors include new analyses comparing antibody levels and T cell magnitude across disease severity levels. (Incidentally, please indicate results of statistical testing for S2D and S2H as in S2C).”

R: As the Reviewer requested, we have added results of statistical testing for Supplementary Figure S2D and Figure S2H.

“2. While this is useful, I was specifically requesting an analysis of how diversity of the TCR repertoire might vary as a function of disease severity (and age and sex). The goal is to determine if age, sex, or disease severity correlate at all with TCR repertoire diversity and/or epitope-specific persistence. I feel that this is important to include regardless of the result of these tests; if for instance the correlation between N clones and stability is different across ages or disease severity, this would be an important finding.”

R: As the Reviewer requested, in the amended version of the paper, we have added Supplementary Figure S4, showing a correlation between proportion of recognized epitopes and disease severity or age or sex and Figure S7A-C, showing a correlation between proportion of clonotypes and disease severity or age or sex.

“3. In addition, the authors' change in response to point 11 is very vague to the point of potential misinterpretation. Please simply report the fraction of these clonotypes that were associated with each epitope.”

R: We have added fraction of each epitope-specific clonotypes in Supplementary Table S4.

“4. Lastly, under "Subhead 13: Data availability", the authors only list availability of previously published and publicly available analytical software. It is vital that the actual data for all of these analyses also be made publicly available upon publication.”

R: We have added Subhead 13: Data availability containing link to GitHub page with actual pipeline and data.

REVIEWERS' COMMENTS:

Reviewer #1 (Remarks to the Author):

The authors have addressed my concerns.